# ATOMIZED DEEP LEARNING MODELS

## ABSTRACT

Deep learning models often tackle the intra-sample structure, such as the order of words in a sentence and pixels in an image, but have not pay much attention to the inter-sample relationship. In this paper, we show that explicitly modeling the inter-sample structure to be more discretized can potentially help model's expressivity. We propose a novel method, Atom Modeling, that can discretize a continuous latent space by drawing an analogy between a data point and an *atom*, which is naturally spaced away from other atoms with distances depending on their intra structures. Specifically, we model each data point as an atom composed of electrons, protons, and neutrons and minimize the potential energy caused by the interatomic force among data points. Through experiments with qualitative analysis in our proposed Atom Modeling on synthetic and real datasets, we find that Atom Modeling can improve the performance by maintaining the inter-sample relation and can capture an interpretable intra-sample relation by mapping each component in a data point to electron/proton/neutron.

## 1 INTRODUCTION

Multiple widely used neural networks are composed of two parts: the first part projects data points into another space, and the other part of the model does further regression/classification upon this space. By transforming raw data features to another potentially more tractable space, deep learning models have recently shown potential in many areas, ranging from dialogue systems (Vinyals & Le, 2015; López et al., 2017; Chen et al., 2017), medical image analysis (Kononenko, 2001; Ker et al., 2017; Erickson et al., 2017; Litjens et al., 2017; Razzak et al., 2018; Bakator & Radosav, 2018) to robotics (Peters et al., 2003; Kober et al., 2013; Pierson & Gashler, 2017; Sünderhauf et al., 2018).

One major challenge in deep learning is to model better intra- and inter-sample structures for complex data features. Recent works often model the *intra-sample* structure by considering the order and adjacency of the input features, for instance, positional encoding for texts/speech in Transformers (Vaswani et al., 2017) and kernel width for images in convolution neural networks (LeCun et al., 2015). Regarding the *inter-sample* structure, literature often assumes that a dataset can be represented in a continuous space and an interpolation of two embeddings might be meaningful (Bowman et al., 2016; Chen et al., 2016), while the data might be naturally discrete (van den Oord et al., 2017). Moreover, the mainstream relies on a non-fully transparent optimization function to reorganize the space. Therefore, in this work, we would like to explore how to explicitly, dynamically rearrange the space (inter-sample structure) by leveraging the intra-sample structures.

Inspired by Atomic Physics, where *atom* is the smallest unit of matter and meanwhile discretely distributed, we propose that we can model a data point as an atom. As illustrated in the left of Figure 2, an atom in a Bohr model (Bohr, 1913), an often adopted concept in Physics (Halliday et al., 2013) and Chemistry (Brown, 2009), contains a dense nucleus, which is composed of the positively charged protons and uncharged neutrons, surrounded by orbiting negatively charged electrons with a *nucleus radius*. Further, multiple atoms can have interatomic forces, composed of attractive and repulsive forces, that make the atoms distant away (non-zero). Such interatomic forces are also the reason for the atoms to form molecules, crystals and metals in our observable life.

In this paper, we propose Atom Modeling, a science- and theoretically-based method that explicitly model the intra-sample relation via atomic structure and the inter-sample relation via interactomic forces. Specifically, we consider a data point as an atom and let a model automatically learn the mapping of each component in a data point to an electron, a proton, or a neutron. We then estimate

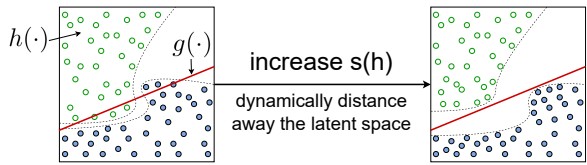

Figure 1: An illustration of our motivation, where a neural network is seen to be composed of two groups of functions $h(\cdot)$ and $g(\cdot)$. For a fixed capacity of $g(\cdot)$, e.g., a linear function, the left latent space is not able to be separated. If the lower left part is distanced away, the same $g(\cdot)$ can split them.

interatomic forces with the learned subatomic particles, nucleus radius and atomic spacing. Finally, the model is optimized to minimize the potential energy induced by the interatomic forces and maintain the balance of total charges and the number of electrons, protons, and neutrons. This method is not only found effective, but also easy to implement in tens of lines for any model architecture.

We validate the effects of Atom Modeling on synthetic data and real data in the domains of text and image classification as well as on convolutional neural networks and transformers. The empirical results show that Atom Modeling can consistently improve the performance accross data amounts, domains, and output complexity. The analyses demonstrate that Atom Modeling can capture intra-sample structures with interpretable meanings of subatomic particles in a data point, while forms an inter-sample structure that increases the model expressivity.

Our contributions are:

- We propose to look into the problem of discrete representation in deep learning models.
- We propose Atom Modeling, a simple method of Atomic Physics for machine learning where the distances among data points (inter-sample structure) naturally depend on their intra-sample structure.
- We empirically demonstrate that Atom Modeling can improve the performance across different setups and provides an interpretable atomic structure of a data point.

## 2 MOTIVATION

We are motivated by a property of the hidden layer of a neural network, and the property of the naturally existing atoms.

### 2.1 DISCRETE REPRESENTATION

A neural network can be represented as a composition of functions $f(\cdot) = f_N \circ f_{N-1} \circ ... \circ f_1(\cdot)$, where each function is one of its $N$ layers. When a neural network $f(\cdot)$ is seen as two groups of functions $f(x) = g \circ h(x)$, the first half $h(\cdot) = f_n \circ f_{n-1} \circ ... \circ f_1(\cdot)$ encodes the input into a hidden space and the second half $g(\cdot) = f_N \circ f_{N-1} \circ ... \circ f_{n+1}(\cdot)$ transforms the latent into the output space.

We consider that in a situation when the model capacity of the second half $c(g)$ is fixed. The projected latent space by the first half functions are hence important to the final output of the model. For easier mathematical description, we denote a quantization of the simplicity of the encoded hidden space as $s(h)$. The whole model capacity $c(f)$ is therefore bounded by:

$$\min(s(h), c(g)) \leq c(f) \leq \max(s(h), c(g)) \tag{1}$$

Our intuition is that if the distance and the shape of two classes are hard to separate, the simplicity $s(h)$ is small. Then the second half functions with a model capacity equal to a linear function cannot separate them. An example is shown in Figure 1. If the points' distances are larger, they can be split by the same linear function. That is, the space simplicity $s(h)$ can be promoted by a more discrete latent space, so one of the bounds of the model capacity $c(f)$ can be improved.

However, if only naively increasing the distances among all the points, the space will be unboundly enlarged and the relative positions might not be changed much. A method that focuses more on separating points that are nearby but with different properties (e.g., the lower left side in the two plots of Figure 1) is desired.

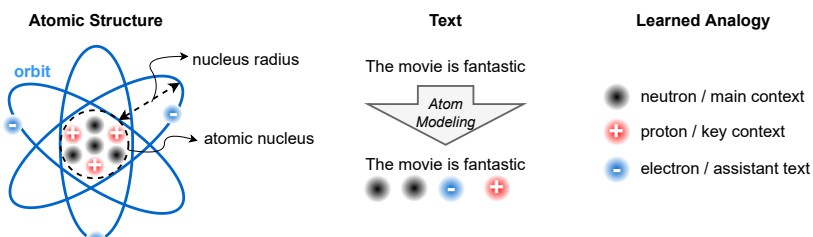

Figure 2: The motivation of Atom Modeling, by which a model will automatically learn a mapping between the atomic structure (left; Bohr model (Bohr, 1913)) and the features of a data point (middle; the tokens in a piece of text). A learned analogy (right) can be that less variant main contexts are like the neutrons; key contexts are like the protons; assistant texts that often co-occur with the protons are like the electrons.

## 2.2 ATOMIC PHYSICS

Atom is an unit in the nature that composes molecules, crystals, metals in the nature and helps determine their properties (Halliday et al., 2013; Brown, 2009). An atom consists three types of subatomic particles: electrons, protons, and neutrons. Among them, an electron has a negative charge, a proton has a positive charge, and a neutron has no charge. In terms of their masses, scientists have empirically measured that an electron has approximately 1/1836 mass of a proton and a proton is slightly lighter than a neutron (Mohr et al., 2008).

The atomic structure of a Bohr model (Bohr, 1913) is shown in Figure 2, where the protons and neutrons form a *nucleus* that occupies a small volume of the atom; the electrons orbit around the nucleus with a *nucleus radius*. Such atoms have two primary types of forces: *intraatomic* and *interatomic* forces. The intraatomic forces include the strong nucleus force that forms protons and neutrons into a nucleus, while the interatomic forces bind two atoms and avoid them having zero distance.

The major motivating property of atoms are that the distance between two atoms is related to their atomic structures, such as the number of the protons and electrons. As shown in Figure 3c, the intuition is that only atoms that are already close to each other and have similar structures, e.g., they are all positively charged, will be distanced away.

This property can be desirable for us to discretize data representation in neural networks. In this work, we (1) assume that there is intraatomic force in the mapping from a data point to the atomic structure and (2) borrow the idea of the interatomic forces to explicitly model the inter-sample structure.

## 3 ATOM MODELING

Our proposed method include two parts: the intra-sample structure modeling and the inter-sample structure regulation.

As illustrated in Figure 2, we regard each component in a data point as a subatomic particle. The mapping from each component to a subatomic particle forms the intra-sample structures, and these intra-sample structures are jointly learned with the dynamic distance function among data points, i.e., the inter-sample structures. Atom modeling has a property to increase the distance among data points with similar intra-sample structures while not affecting data points with different intra-sample structures. We provide this proof in the Appendix.

## 3.1 INTRA-SAMPLE STRUCTURE

A simple atomic structure (Bohr, 1913) includes three subatomic particles, protons, electrons, and neutrons. This structure has several properties, such as *charges*, *masses*, *nucleus radius* and the *distances between two particles*. We will show that these properties can be simply determined by a learnable *charge* associated with each component in a data point.

**Charges.**  We assign a *charge*, $q \in \mathbb{R}$, to each component of a data point. We set the range of $q$ to be $[-1, 1] \subseteq \mathbb{R}$ since in nature, the charge of the subatomic particles, electron, neutron, and proton, are respectively $-1$, $0$, and $+1$. To obtain $q$ in a neural network, we first assume a model encodes each component of a data point into an $h$-dimensional embedding $\mathbf{e} \in \mathbb{R}^h$ in one of its hidden layer (the $n$-th layer in section 2.1). Then without additional parameters, we can transform arbitrary one dimension in $\mathbf{e}$, which is denoted as $e^q \in \mathbb{R}$, into the charge by $q = 2\sigma(e^q) - 1$, where $\sigma(\cdot)$ indicates the Sigmoid function.

**Masses.**  The empirical results in Atomic Physics show that electrons have limited masses while protons and neutrons masses are in a similar level (Mohr et al., 2008). Following these results, we approximate the mass $m \in \mathbb{R}$ of each component in a data point as shown in Figure 3a, i.e., the mass is about one when the charge is larger than or equal to 0; the mass linearly reduced to zero when the charge is negative. Our analogy of mass is mathematically defined as $m = 1 - \max(-q, 0)$.

**Nucleus.**  An atomic nucleus is composed of protons and neutrons, which occupy nearly all the mass of an atom. Therefore, we approximate the position of the atomic nucleus by weighted average of each subatomic particle with its mass as the weight. The position of the atomic nucleus $\boldsymbol{\mu}^{\mathbf{P}} \in \mathbb{R}^{h-1}$ is then formulated as $\boldsymbol{\mu}^{\mathbf{P}} = \frac{1}{|\mathcal{A}|} \Sigma_{i \in \mathcal{A}} \mathbf{e}^{\mathbf{P}}{}_i m_i$, where $i$ is the $i$-th component of a data point (atom) $\mathcal{A}$, and $|\mathcal{A}|$ is the number of components. $\mathbf{e}^{\mathbf{P}}$ is the position of each component in the embedding space. Without additional parameters, we take the remaining dimensions of $\mathbf{e}$ (without $e^q$) as the position in the embedding space of each component in a data point, so $\mathbf{e}^{\mathbf{P}} \in \mathbb{R}^{h-1}$.

**Nucleus Radius.**  Since the outermost particles of an atom are electrons, we heuristically approximate the nucleus radius as the average distance from electrons to the nucleus. The nucleus radius $r \in \mathbb{R}$ is hence formulated as $r = \frac{1}{|\mathcal{A}|} \sum_{i \in \mathcal{A}} ||\mathbf{e}^{\mathbf{P}}{}_i(1 - m_i) - \boldsymbol{\mu}^{\mathbf{P}}||_p$, where $|| \cdot ||_p$ denotes $p$-norm.

**Distances.**  Considering the geometry of atomic structure, as illustrated in Figure 3b, we approximate the distances between a pair of particles in different atoms (atoms $\mathcal{A}_1$, $\mathcal{A}_2$) by $d_{ij} = ||\boldsymbol{\mu}^{\mathbf{P}}{}_1 - \boldsymbol{\mu}^{\mathbf{P}}{}_2||_p$ if they are *homoelectricity* ($q_i q_j > 0$, where $i \in \mathcal{A}_1$ and $j \in \mathcal{A}_2$). If they are *heteroelectricity* ($q_i q_j < 0$), the distance is approximated by $d_{ij} = ||\boldsymbol{\mu}^{\mathbf{P}}{}_1 - \boldsymbol{\mu}^{\mathbf{P}}{}_2||_p + (r_1 + r_2)/2$. The nucleus position $\boldsymbol{\mu}^{\mathbf{P}}$ and nucleus radius $r$ with subscripts 1 and 2 corresponds to atoms $\mathcal{A}_1$ and $\mathcal{A}_2$ respectively.

**Balance of Charges and Number of Particles.**  Moreover, we follow the findings in Atomic Physics that an atom tends to be electrically neutral for stability. Meanwhile, the number of neutrons is usually about the same as the number of protons. Therefore, we propose two losses, $L_{charge}$ and $L_{neutrons}$, to regularize the charges and number of neutrons.

$$L_{charge} = (\sum_{i \in \mathcal{A}} q)^2,$$
$$L_{neutrons} = (\sum_{i \in \mathcal{A}} q^2 - \frac{2}{3}|\mathcal{A}|)^2. \tag{2}$$

The idea is that $L_{charge}$ is the mean square error between the total charge and zero, and $L_{neutrons}$ is an approximation of the total number of charged particles (protons and electrons; $\Sigma_{i \in \mathcal{A}} q^2$). The charged particles should occupy about 2/3 of the total number of particles in an atom ($|\mathcal{A}|$). That is, the remaining 1/3 is the number of neutrons.

## 3.2 INTER-SAMPLE STRUCTURE

After assigning a charge for each component in a data point, we can take advantage of the derived characteristics, i.e., mass, nucleus, radius, and distance, to compute the **interatomic forces** for building the inter-sample structure.

As illustrated in Figure 3c, a widely accepted understanding is that the interatomic force is composed of two parts: *repulsive force*, which pushes away two homoelectric particles, and *cohesive force*, which pulls closer two heteroeletric particles. The combination of the two forces will compose a curve of their potential energy with respect to the atom spacing. Such curve has a *Balance Point* which results in the lowest potential energy and often locates at a short distance between two atoms but

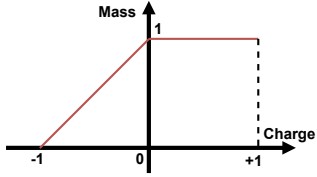

(a) The approximated masses of charges.

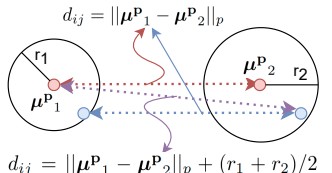

(b) The approximated distances of particles.

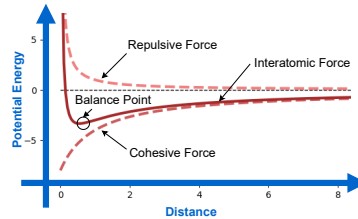

(c) The effect of interatomic force.

Figure 3: The atomic analogy of (a) masses and (b) forces; (c) a typical resulting potential energy curve with respect to the distance among two atoms induced by the interatomic force, which is a combination of repulsive and cohesive forces.

must be larger than zero. When distance is closer to zero, the potential energy increases dramatically, thus naturally preventing two atoms become "continuous" (with zero distance).

In Atom Modeling, we compute the potential energy caused by Columb Forces (Halliday et al., 2013) for randomly sampled particle pairs drawn from two data points ($\mathcal{A}_1$, $\mathcal{A}_2$) and take the summation. That is, we aim to minimize the loss defined as:

$$L_f = \sum_{i \sim \mathcal{A}_1, j \sim \mathcal{A}_2}^{\min(|\mathcal{A}_1|, |\mathcal{A}_2|)} \frac{q_i q_j}{d_{ij}} . \tag{3}$$

During implementation, for computational efficiency, we randomly pair components from data points in the same batch to compute $L_f$. Therefore, the order of complexity does not increase.

### 3.3 OPTIMIZATION

During training, we take a weighted sum of the three losses, $L_f, L_{charge}$, and $L_{neutrons}$, and add them to the original loss function $L_{ori}$, e.g., a cross-entropy loss. The model will then be trained by minimizing the following loss function:

$$\mathcal{L} = L_{ori} + c_f L_f + c_{charge} L_{charge} + c_{neutrons} L_{neutrons} . \tag{4}$$

We empirically observe that the coefficients $c_f$, $c_{charge}$, and $c_{neutrons}$ do not require careful tuning and can often be the same number.

### 3.4 THEORETICAL RESULTS

**Theorem 3.1.** *Data points with similar intra-sample structures can have longer balance distance; ones with more opposite intra-sample structures can have shorter balance distance. Specifically, the balance distance is proportional to $\frac{\sqrt{k}+1}{k-1}$, where $k$ indicates the difference between the intra-sample structures of two data points and satisfies $k > 1$.*

We provide this proof in the Appendix.

## 4 EXPERIMENTS

In order to validate the effects of Atom Modeling, we conduct extensive experiments across multiple datasets on synthetic, text (Warstadt et al., 2019; Sheng & Uthus, 2020; Gräßer et al., 2018) and image domains (Parkhi et al., 2012; Nilsback & Zisserman, 2008; Deng et al., 2009) with two model architectures, transformer (Vaswani et al., 2017; Devlin et al., 2019) and convolution neural network (He et al., 2016). We compare our method with two commonly-seen distance functions, p-norm with $p = 1$ and $p = 2$, as our designed baselines, since to the best of our knowledge, we have not seen prior works discuss this type of method. We also answer the question: "*what are the learned intra- and inter-structures?*" by qualitative analyses in the next section.

### 4.1 SYNTHETIC EXPERIMENT

To gain more insights of Atom Modeling, we design the following synthetic experiment. In this way, we can visualize the inter-sample structure in the latent space without further projection, which can change the structure.

**Setup - Data Generation.** Inspired by the complexity of real data, e.g., texts and images, we propose a synthetic dataset where each data point has multiple components. We design each data point to contain five 2-dimensional vectors, where each vector (input feature) is sampled from a Gaussian Mixture Model composed of two Gaussian distributions, $c_0$ and $c_1$. If most of the input features in a data point is sampled from $c_0$, we label the data point as 0; otherwise, we label it as 1. The generated dataset is plotted in Figure 4.

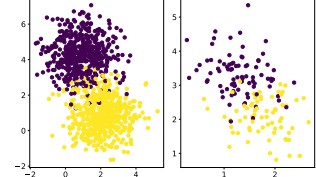

Figure 4: The synthetic dataset. The left is the two Gaussian distributions where input features are sampled from; each point in the right is the average of the five input features in each data point.

**Setup - Model Architecture.** For fair comparison and easy visualization, all four methods are trained on the same neural network with two linear layers. The first layer (2x3) is fed with the data points of size (5x2), and its output will be of size (5x3) in a hidden space. We then take the first dimension of this output as the weight (5x1) and the rest as the embeddings of the five input features (5x2). Hence, we can compute the weighted sum of the five embeddings and obtain a two-dimensional vector. We take this vector as the embedding of a data point and feed it into the second layer (2x2) to classify the data points as 0 or 1.

**Results.** As shown in Figure 5, across 10 random runs, the same model trained with only cross-entropy loss achieves an average 87% accuracy and has a relatively low variance. With the addition of increasing 1-norm or 2-norm distance among data points in the latent space, the highest accuracy can achieve is about 90% accuracy. However, since the variances are simultaneously increased, the average accuracy is only enhanced a bit to 87%-88%. If we apply Atom Modeling, the average accuracy is significantly improved to 92% and with a median 96%. Overall, we observe that distancing away data point in the latent space can enhance the chance to improve the model performance. However, 1-norm and 2-norm distances treat every data points the same, hence the learned data point positions in the latent space can be relatively the same and only the scale is enlarged. In the other hand, Atom Modeling does not treat every data points equally but focuses on increasing the data points that should not be too closed, e.g., points in the neighborhood but in different classes.

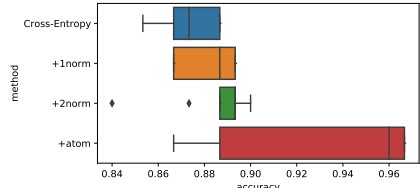

Figure 5: The testing results of four different training methods.

### 4.2 TEXT CLASSIFICATION

We validate the effect of Atom Modeling on text classification, which is an often seen real application where the data points are composed of multiple, ordered tokens. The number of tokens in data points are various.

**Setup.** We conduct experiments on three datasets with different scale and number of classes, CoLA (Wang et al., 2018; Warstadt et al., 2019), Poem (Sheng & Uthus, 2020), and Drugs (Gräßer et al., 2018). CoLA is a set of about 10k sentences annotated with 2 classes about whether it is a grammatical English sentence based on Linguistic theory. Poem is a corpus about 1k examples for sentiment analysis of classic poems with verses annotated with four classes: negative, no impact, positive, or mixed (both negative and positive). Drugs is a set of about 100k pieces of reviews and each is diagnosed to be one of 50 conditions, e.g., "depression","pain","acne".

We fine-tune a pretrained language model, BERT-base-uncased (Devlin et al., 2019), with cross-entropy loss. To implement 1-norm/2-norm distance regulation on BERT, we take the mean of the n-th layer hidden states of all tokens as the embedding of the data point. To atomize the n-th hidden

| | CoLA | | Poem | | Drugs | |
|---|---|---|---|---|---|---|
| | Acc. | MCC | Acc. | F1 | Acc. | F1 |
| BERT | 83.6±0.4 | 60.0±1.0 | 86.7±2.5 | 60.3±4.5 | 83.5±0.3 | 79.5±0.2 |
| +1-norm | 83.6±0.3 | 59.8±0.8 | 87.3±3.1 | 62.0±3.2 | 80.8±1.3 | 75.7±2.2 |
| +2-norm | 83.7±0.4 | 60.1±0.9 | 87.3±4.0 | 61.8±3.5 | 84.5±0.2 | **80.7±0.3** |
| +Atom | **84.1±0.5** | **61.3±1.3** | **88.6±1.9** | **62.7±2.4** | **84.7±0.4** | 80.5±0.5 |

Table 1: Results of text classification. We report the mean and the standard deviation (the value after ±) of each evaluation metric for three random runs.

layer of the BERT model ($n = 1$ here), we utilize the first dimension in the latent space of each token as its charge $e^q$. Therefore 1-norm/2-norm/Atom Modeling are implemented upon BERT without extra parameters.

**Results.** In Table 1, similar to the results of synthetic experiment, we observe that in most cases Atom Modeling achieves the best results. The method 2-norm distance has a more on-par performance with Atom Modeling on the Drugs dataset. Oppositely, the 1-norm distance method performs similarly or slightly better on CoLA and Poem, but worse than only using cross-entropy on Drugs. One difference from the synthetic experiment is that the utilized transformer architecture does not directly operates on the data points embeddings, but on the components embeddings. In this way, the benefits of Atom Modeling might not be fully leveraged. However, we can still observe the improvements via the jointly learned components embeddings across different number of classes and dataset scales.

### 4.3 IMAGE CLASSIFICATION

We evaluate Atom Modeling on image classification, which is an often seen real application where the data points are composed of pixels with specific arrangement.

**Setup.** We conduct experiments on three datasets with different scale and number of classes, Oxford-Pets (Parkhi et al., 2012), Oxford-Flowers102 (Nilsback & Zisserman, 2008), and ImageNet (Deng et al., 2009). Oxford-IIIT Pets consists of 37 cats and dogs species with roughly 200 images for each class. Oxford Flowers-102 consists of 102 flower categories. Each class consists of between 40 and 258 images. ImageNet consists of about 1M images in 1000 classes. We resize all the images to 32x32 and apply the same data preprocessing method for these three datasets. Note that the resizing will make the tasks more challenging since some details in a image might disappear.

Based on previous implementation of ResNet18 (He et al., 2016), we first concatenate the output channels of the first convolution layer and take each corresponding position of a pixel as its embedding **e**. To implement 1-norm/2-norm distance regulation, we take the mean of the pixel embeddings as the embedding of the data point. To atomize this layer in ResNet, we utilize the first dimension of each embedding as its charge $e^q$. Therefore 1-norm/2-norm/Atom Modeling are implemented upon ResNet without extra parameters.

**Results.** As listed in Table 2, similar to the results of both the synthetic experiment and text classification, we observe that Atom Modeling achieves the highest accuracy in all cases. In most cases, the 2-norm distance method slightly improves the results trained with only cross-entropy loss. However, the 1-norm distance method performs poorly in image classification. We also found that the performance improvements are larger on Oxford-IIIT Pets and Oxford Flowers-102. We conjecture that this is because these two datasets are more fine-grained classification tasks, since some categories of pets and flowers are not easy to distinguish.

### 5 QUANTITATIVE AND QUALITATIVE ANALYSES

As our motivations about the discrete representation and the property of atomic physics, we are curious about (1) the impact of the atomized $n$-th layer, (2) the learned latent space/inter-sample

|  | Oxford-IIIT Pet | | Oxford Flowers-102 | | ImageNet |
|  | dev | test | dev | test |  |
|---|---|---|---|---|---|
| ResNet | 21.58±0.99 | 20.98±1.11 | 59.15±0.98 | 56.73±1.52 | 48.07±0.09 |
| +1-norm | 20.41±1.40 | 19.98±1.09 | 58.20±1.42 | 54.67±2.16 | 47.76±0.20 |
| +2-norm | 22.75±0.41 | 22.39±0.50 | 60.72±0.54 | 58.01±0.82 | 47.90±0.20 |
| +Atom | **23.41±0.57** | **22.53±0.87** | **62.78±1.72** | **59.08±1.18** | **48.23±0.09** |

Table 2: Results of image classification. We report the mean and the standard deviation (the value after ±) of each evaluation metric for three random runs.

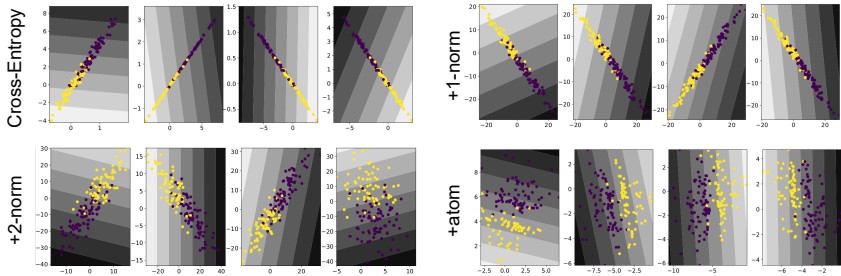

Figure 7: The visualization of the learned inter-sample structure of the synthetic data.

structure, and (3) the learned mapping from data point components to subatomic particles/intra-sample structures.

## 5.1 ATOMIZED THE $n$-TH HIDDEN LAYER

The $n$-th layer (in Section 2.1) split a model into the first part that projects data point into the latent space, and the second part that transforms the latent space to the output. Due to the goal of Atom Modeling to reorganize the latent space for the second part of the model can do better transformation, which hidden layer is reorganized might affect the final results. In Figure 6, which is to apply Atom Modeling in different layer of BERT model on CoLA, we observe that discretizing the first layer generally has the best performance. Recall that we were using a complex model architecture, which does not directly operate on the data point embedding, but the components embeddings. We conjecture that this complexity of the model architecture requires a larger capacity of the second part of the model.

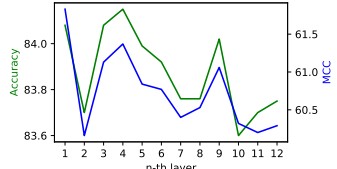

Figure 6: The results of atomizing different hidden layer in a BERT model on CoLA.

## 5.2 VISUALIZATION OF LEARNED INTER-SAMPLE STRUCTURE

After training with different methods, the projection of the same data points to the same latent space will have different relative positions. As plotted in Figure 7, we visualized four random runs of the learned inter-sample structures of models trained with only cross-entropy loss, 1-norm/2-norm distance, and Atom Modeling on the synthetic experiment. We found that training with only the cross-entropy loss results in the embeddings entangled in the middle and has a long and thin shape. When applying additional 1-norm distance, the shape is similar to cross-entropy but the scale of both axes is enlarged about five times. When using 2-norm distance, the shape is more similar to the original dataset. However, there are still multiple overlaps and the scale is enlarged up to ten times. In the end, we found that Atom Modeling not only spreads out the distribution to be like the original dataset, but also distance away middle ones as our motivation. Meanwhile, the scale is remained about the same as the scale learned by cross-entropy.

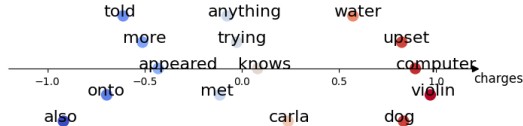

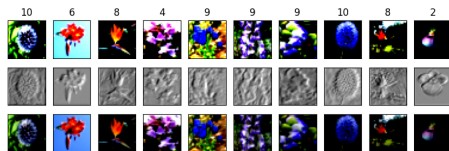

Figure 8: The charges of sampled words in CoLA. The y-axis values are randomly generated. Most key contents are automatically learned to be protons, assistant texts are learned to be electrons.

Figure 9: The charges (second row) of figures in Oxford Flowers-102. The first row and the third row are the original images and the ones multiplied with charges respectively.

### 5.3 Visualization of Learned Intra-Sample Structures

In Atom Modeling, the inter-sample structure depends on the learned intra-sample structure. We look into the learned intra-sample structure in texts and images to have an understanding of how the charges distributed in real world among words and pixels. For texts, we first extract the charges of a token with different context and then take an average. We plot the charges of randomly sampled 15 words of CoLA in Figure 8. We found that in both CoLA and STSB datasets, most often seen words such as "more" and "also" are learned to be electrons, while protons are often Nouns with specific meanings, such as "water" and "violin". This phenomenon validates our expectation that the key contexts are mapped to protons and assistant texts that often co-appear with the key contexts are mapped to electrons. For images, the charges of every pixels are plotted in Figure 9. We observe that the charge distribution is often like a relief sculpture, where electrons are mapped to shadows on object edges and protons are the crucial parts for a model to classify the image. Moreover, we found that the charge distribution sometimes recover some missing details in the processed images. For example, in the right most of Figure 9, the charge distribution draws the other petals that originally hidden in the dark.

## 6 Related Work and Discussion

Recent works have tried to connect Physics with machine learning and data science (Raissi et al., 2017; Karpatne et al., 2017; Bar-Sinai et al., 2019; Raissi et al., 2019; Sitzmann et al., 2020; Krishnapriyan et al., 2021; Giladi et al., 2021; Takeishi & Kalousis, 2021). For computer vision, Hasani et al. (2021) and Ding et al. (2021) model visual reasoning by considering Physical equations such as velocity. Tang et al. (2021) proposes to split images into amplitude and phase as waves to improve the MLP architecture. For natural language processing, there is a surge of researches on the utility of Quantum Physics in texts (Wu et al., 2021). For example, Zhang et al. (2020) applies Quantum Physics to do sentiment analysis for emotions in dialogues. Atom Modeling also explores the potential impact from Nature science to machine learning. However, the goal of Atom Modeling is to approach the question: *"how to promote the discrete nature of data points in a deep learning model?"*

Some will find similarity between Atom Modeling and regularization (Tibshirani, 1996; Nie et al., 2010; Gulrajani et al., 2017) and kernel methods (Muller et al., 2001; Keerthi & Lin, 2003; Hofmann et al., 2008). The main differences from them are that first Atom Modeling constrains the underlying data representation rather than the model parameters. Second, Atom Modeling aims to enhance the expressivity of the inter-sample structure in a dataset and specifically model the discrete nature in an unchanged dimensional space, instead of mapping from lower to higher dimensional space.

## 7 Conclusion

In this paper, we propose Atom Modeling, a science- and theory-based approach to promote a discrete representation in a deep learning model. Moreover, by drawing an analogy between data and subatomic components, Atom Modeling learns interpretable intra-sample structure for each data point. An important property is that a model trained with Atom Modeling learns an inter-sample structure (distances among data points) depending on the intra-sample structure. Leveraging this property can potentially improve model capacity. Moving forward, we see that Atom Modeling provides a possible way to improve the current neural network and hope it provides a base for future works to consider more about the discrete nature.

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

# A    PROOF OF THEOREM 3.1

$$L_f = \sum_{i \in \mathcal{A}_1, i \in \mathcal{A}_2}^{\mathcal{A}_1 \mathcal{A}_2} \frac{q_i q_j}{d_{ij}}$$
$$= \sum_{q_i q_j > 0} \frac{q_i q_j}{\|\mu_1 - \mu_2\|_p} + \sum_{q_i q_j < 0} \frac{q_i q_j}{\|\mu_1 - \mu_2\|_p + \frac{r_1 + r_2}{2}} \tag{5}$$

Set $d_{12} = \|\mu_1 - \mu_2\|_p$ and $\tilde{r} = \frac{r_1 + r_2}{2}$.

$$\frac{\partial L_f}{\partial d_{12}} = \frac{\sum_{q_i q_j > 0} -q_i q_j (d + \tilde{r})^2 + \sum_{q_i q_j < 0} -q_i q_j d^2}{d^2 (d + \tilde{r})^2} = 0 \tag{6}$$

Set $c_1 = \sum_{q_i q_j > 0} q_i q_j$ and $c_2 = \sum_{q_i q_j < 0} -q_i q_j$.

$$\sum_{q_i q_j > 0} -q_i q_j (d + \tilde{r})^2 + \sum_{q_i q_j < 0} -q_i q_j d^2$$
$$= (-c_1 + c_2) d^2 - 2 c_1 d \tilde{r} - c_1 \tilde{r}^2 \tag{7}$$
$$= (-c_1 + c_2)(d - \frac{c_1 \tilde{r}}{-c_1 + c_2})^2 - (-c_1 + c_2)(\frac{c_1 \tilde{r}}{-c_1 + c_2})^2 - c_1 \tilde{r}^2 = 0$$

$$(d - \frac{c_1 \tilde{r}}{-c_1 + c_2})^2 = \frac{c_1^2 + c_1(-c_1 + c_2)}{(-c_1 + c_2)^2} \tilde{r}^2 = \frac{c_1 c_2}{(-c_1 + c_2)^2} \tilde{r}^2 \tag{8}$$

$$d = (\frac{c_1}{-c_1 + c_2} \pm \sqrt{\frac{c_1 c_2}{(-c_1 + c_2)^2}}) \tilde{r} \tag{9}$$

If $c_1 = c_2$, the right hand side divided by zero is undefined.

If $c_1 > c_2$ and recall that $d = \|\cdot\|_p \geq 0$, then $\frac{c_1}{c_1 - c_2} \leq \sqrt{\frac{c_1 c_2}{(-c_1 + c_2)^2}}$. This results in $c_1^2 < c_1 c_2$, contradict to the premise $c_1 > c_2$.

This equation can be satisfied only when $c_1 < c_2$. A balance point exists in $d = \frac{c_1 + \sqrt{c_1 c_2}}{c_2 - c_1} \tilde{r}$.

Suppose that $c_2 = k c_1$ with $k > 1$, then,

$$d = (\frac{c_1 + \sqrt{c_1 c_2}}{c_2 - c_1} \tilde{r}$$
$$= \frac{c_1 + \sqrt{k} c_1}{(k - 1) c_1} \tilde{r} = \frac{\sqrt{k} + 1}{k - 1} c_1 \tilde{r} \tag{10}$$

which is monotonically decreasing given $k > 1$.

