# OpenReview forum: "Atomized Deep Learning Models"
_ICLR.cc/2023/Conference — Submitted to ICLR 2023_

### Official Review · Reviewer_vXfj · 2022-10-24

**Confidence:** 3
**Correctness:** 2
**Technical Novelty And Significance:** 2
**Empirical Novelty And Significance:** 2
**Recommendation:** 3

**Clarity, Quality, Novelty And Reproducibility:**

Given the originality of the approach, the presentation is relatively clear. The results also seems reproducible.

The closest works that I can think of would be other methods trying to enforce some specific structure in the latent space such as hyperbolic representation learning [1] or equivarient representation learning [2].

----------

[1] Maximilian Nickel, Douwe Kiela
Poincaré Embeddings for Learning Hierarchical Representations

[2] Sara Sabour, Nicholas Frosst, Geoffrey E. Hinton:
Dynamic Routing Between Capsules. NeurIPS 2017

**Strength And Weaknesses:**

Strengths:
- The learned embeddings seem to have interesting properties. It could be interesting to embed sets and discover their inner structure.
- Experiments have been conducted in very different settings in terms of data (toy dataset, image classification, text classification) or architecture (MLP, ConvNet, Transformer)
- The descriptions are clear and straightforward

Weaknesses:
- One important point that might not have enough emphasis is that the model is targeted to data that could be represented by a set of embeddings. Images as collections of pixel embeddings, or sentences as a set of word embeddings. Making it clear early on would help readability.
- The motivation is also not really convincing. The main goal seems to be that embeddings shouldn't lie too close one to another so that a decision frontier is easy to find. I would question the soundness of the assumption (what about generalization then for instance?), but even when taking it at face value, the proposed solution seems like a very contrived and unpractical way to solve this specific problem.
- Given the peculiar constraint on the latent space, its analysis is insufficient. We would need more than a few qualitative samples to understand what is happening. For instance, what kind of properties of the data do they hope will be captured in the intra-atom structure, and conversely, in the inter-atom relationship? We can infer some answers from the experiments, but they need to be set beforehand and explained in the introduction, and then explicited in the results.
- Why are the image experiments performed in 32x32? It seems far to small in for any reasonnable benchmark.
- Many geometrical properties and emergent spatial structures work in a 3-dimensional space and but break in higher dimensions. However, in the proposed model, the atoms are in a large (h-1)-dimensional space. The illustrative experiments they perform are in 2-D space as well. Have the authors considered this limit to their analogy?

**Summary Of The Paper:**

The paper proposes to structure the latent space of a deep neural network following the Bohr model of atoms. In this latent space, each datapoint is to be embedded as an 'atom', i.e. a set of particles represented by their position and their charge.
Using these values, the authors can then compute losses to make sure the atoms respect physical constraints. Inside an atom, the sum of the charges of all particles is pushed towards 0, while the sum of squared charges is pushed towards 2/3 of the number of particles. Moreover, in this latent space, pairs of atoms are moved so that they minimize their potential energy, making sure that atoms of similar structures cannot stay too close to each other.
These losses are added to the original classification loss to solve the task.


**Summary Of The Review:**

The proposed approach is suprising and interesting. It could be very well suited to embed sets and to discover structure within those sets.
However, it is also difficult to build an intuition of its inner workings from the provided results.

Considering how complex and difficult to analyze the proposed approach is, it needs a strong motivation to justify, as well as a extensive evaluation of the properties we can find in the latent space.

---

### Official Review · Reviewer_vH4E · 2022-10-28

**Confidence:** 3
**Correctness:** 2
**Technical Novelty And Significance:** 4
**Empirical Novelty And Significance:** 2
**Recommendation:** 3

**Clarity, Quality, Novelty And Reproducibility:**

Clarity & Quality:
- Terms are not entirely clear. What are "model capacity" and "quantization of the simplicity", and what do they mean? The symbols `c` and `s` presumably are capacity and simplicity, but that is never mentioned. Why is the model capacity of `f` bounded in the way described? Section 2.1 is generally hard to understand, and since there are no references I can't tell if these are newly introduced terms or I should refer to some prior work.
- There are many approximations that are not fully explained or justified (e.g. most points in section 3.1). It would have been good to validate those approximations by looking at their values in final models.
- The grammar and writing could be improved. Given that this paper introduces novel terminology, it seems especially important to have clear writing. E.g. "However, if only naively increasing the distances among all the points, the space will be unboundly enlarged and the relative positions might not be changed much."

Novelty:
Approaching representation learning in this way seems very novel.

Reproducibility:
There are many small details involved in implementing this technique that do not seem to be appropriately described. It would be useful to expand on these, either releasing the code itself (which would be ideal), or including this in the appendix. For example, the method as mentioned in the paper can be implemented in tens of lines for any model architecture, so it would be good to include those tens of lines within the paper itself.

**Strength And Weaknesses:**

Strengths:
- This is a very novel and creative idea. Applying inspiration from physics to machine learning can lead to impactful methods.
- The method does not seem to add much complexity or overhead to the training process (though this appears to not be explicitly evaluated).

Weaknesses:
- The results seem fairly unconvincing. The baselines being compared to appear to have been implemented by the authors themselves, and the dataset choices seem odd. It would be better to start with an externally published and tuned baseline, implement this paper's proposed losses on those models, evaluate on the same datasets, and compare the results. Doing this process for both a recent image and text based model (e.g. ALBERT) seems appropriate.
- The theoretical grounding and justification for this approach seem shaky and could be made more clear. It's not obvious why all the additional structure around atomic modeling is more useful than other ways of affecting the representation space (for example, disentangled representation learning or contrastive losses). This should also ideally be compared to other modern representation learning techniques. Combined with the results not having externally validated baselines, it's hard to understand the significance of the proposed novelty.
- The clarity of the paper could be improved significantly, especially since this paper introduces many equations and approximations. See comments in the section below.

**Summary Of The Paper:**

This paper introduces additional loss terms that impose structure on the representation networks learn. The definition of those terms is inspired by atom and interactions between atoms (masses, charges, etc.), and can loosely be thought of as regularizing the representation space by separating the embeddings.

The authors evaluate their method on several datasets, and show that their approach leads to more meaningful embeddings.

**Summary Of The Review:**

Overall, the paper presents a novel and intriguing method for learning relationships within samples and between samples, but the methodology of the paper is unclear in many places, the theoretical grounding is not adequately justified, and the results are not clearly significant enough to demonstrate the effectiveness of the method as a whole.

---

### Official Review · Reviewer_hSD9 · 2022-10-28

**Confidence:** 3
**Clarity, Quality, Novelty And Reproducibility:** Please see the Weaknesses
**Correctness:** 3
**Technical Novelty And Significance:** 2
**Empirical Novelty And Significance:** Not applicable
**Recommendation:** 3

**Strength And Weaknesses:**

Strength

1 Bringing the concepts of atoms into the neural network is interesting.

Weaknesses

1 The setup of some experiments is strange. For example, the image from ImageNet is resized to 32x32. It does not make much sense since it will make the input much noise and make the comparison unconvincing.

2 The writing can be improved a lot. Many parts of the manuscript are unclear. E.g.
- What’s the meaning of $\circ$ in the first line of section 2.1?

- What's the meaning of the simplicity of a model?

- The meaning of k is unclear. What's the meaning of "difference between the intra-sample structures of two data point"?

3 The evaluations are quite insufficient.
- There is no sensitiveness analysis with respect to the three weighting factors of the losses, although the authors claim that they have little influence on the performance.

- There is no comparison with the optimization method along the line of contrastive learning which can also reduce the intra-class distance and enlarge the inter-class distance. The baseline models are too weak to show the effectiveness of the proposed method.

4 There are many related works missing, making the novelty of the proposed method unclear. For example, the regularization of the latent space is quite standard in the field of disentanglement learning, auto-encoder, and contrastive learning. It is unclear what's the difference between the proposed method and these related works.

**Summary Of The Paper:**

This manuscript maps some concepts of the Atom to the hidden space of a hidden output of a NN. The hidden features are transferred to the concept of Atom and regularized with charge, neutrons, and forces as an additional loss. The proposed method is evaluated on several machine learning tasks with respect to some naive baselines.

**Summary Of The Review:**

The main concerns are the insufficient evaluation (baselines and weird datasets) and the novelty of the proposed method with respect to many other similar works (disentanglement learning, auto-encoder, and contrastive learning).

---

### Official Review · Reviewer_qaGR · 2022-10-29

**Confidence:** 3
**Correctness:** 2
**Technical Novelty And Significance:** 2
**Empirical Novelty And Significance:** 1
**Recommendation:** 3

**Clarity, Quality, Novelty And Reproducibility:**

This paper is written in fair quality but does not meet the standard of ICLR. I have provided my evaluations in the answer to the previous question.

**Strength And Weaknesses:**

**Strengh**

1. The paper is generally well-organized, and clearly presented

2. The background of atomic physics is concise and helpful.


**Weakness and questions**

1.  In the abstract, the claim '*have not pay much attention to the inter-sample relationship*' seems even wrong to me. For example, research on graph neural networks is mainly on sample-level representations. And there is extensive research on regularizing the deep feature space based on intra-sample or inter-class properties.

2. *we show that explicitly modeling the intersample structure to be more discretized can potentially help model’s expressivity*. I cannot see a very clear motivation for why a more discretized feature space can help improve model expressiveness. My main concern is based on the fact that well atomic physics is a well-established research, it does not necessarily apply to all data modalities that deep neural networks are dealing with. And I fail to find sufficient discussions on the motivation behind extending atomic physics to data we use in deep learning. In short, I do not see a clear connection between atom-level physics and modalities like images and text.

3. The previous concern is also confirmed by the experimental results. In my opinion, most of the performance improvements on all the real-world datasets cannot be considered significant.

4. This paper fails to position itself under a more general context. The method is only compared against some simple alternatives like 1-norm 2-norm. There are potentially more studies on feature geometry that can be included. For example, center loss [1] and orthogonal low-rank geometry [2].

5. The experiment settings, such as 32*32 ImageNet classification are not standard. If it is due to hardware constraints, then standard low-resolution experiments like CIFAR-10 and CIFAR-100 can be more convincing.

6. I personally do not recommend calling this method *science- and theory-based*. It is a science-inspired method. But the theory provided in this paper does not support the effectiveness. For example, the proof in Theorem 3.1 describes the mathematical properties of the new regularization term, but does not explain why it should work on real-world data.

[1] A Discriminative Feature Learning Approach for Deep Face Recognition, ECCV 2016

[2] OLE: Orthogonal Low-rank Embedding, A Plug and Play Geometric Loss for Deep Learning, CVPR 2018

**Summary Of The Paper:**

This paper introduces a very interesting view of deep latent space. The authors introduce a idea inspired by phisics, and propose a new model regularization method. The new regularization method promotes distance between data points based on they intra-sample properties learned from the task. The idea is validated on synthesized toy experiments and real-world classification task.

**Summary Of The Review:**

My main concerns are:

1. This paper fails to position itself under a more general context. Therefore it is hard to evaluate the contribution.

2. The motivations behind extending atomic physics to real-world data modalities remain unclear to me.

3. Experiments are weak.

---

### Official Review · Reviewer_KJc9 · 2022-11-04

**Confidence:** 3
**Correctness:** 3
**Technical Novelty And Significance:** 3
**Empirical Novelty And Significance:** 4
**Recommendation:** 3

**Clarity, Quality, Novelty And Reproducibility:**

Using Atom Modeling for inter-sample interaction is novel. I think the main results of the paper are reproducible, but there may be a mismatch with mainstream settings on image classification benchmarks.

**Strength And Weaknesses:**

**[Strength]**

- The modeling of feature interactions from an atomic perspective is novel.
- Considering cross-sample interactions complements the current single-input-single-output mainstream neural networks.
- The proposed module has been generally validated on different models and datasets.

**[Weaknesses]**

- The discussion of related work is very inadequate. There are many existing fields that consider inter-sample feature interactions (e.g., contrastive learning), but the authors do not expand on them.
- Figure 1 shows a good motivation -- considering the distance relationship between **different sample points** makes sense. This drives us to implement feature interactions across data samples. However, using Atom Modeling to do this is not as intuitive and, for example, not as clean as Supervised Contrastive Learning [R1]. I don't think Atom Modeling is well motivated from Figure 1, and the description of this section may need to be rewritten.
- The experimental configuration on image classification does not match the mainstream research. In most recent work evaluated on Pets, Flowers or ImageNet datasets, the image resolution is typically set to 224 and their performance far exceeds that of this paper. These facts make the conclusions of this work on the image classification task less convincing.

-----------
[R1] Khosla, Prannay, et al. "Supervised contrastive learning." Advances in Neural Information Processing Systems 33 (2020): 18661-18673.


**Summary Of The Paper:**

This paper post-processes the representation of deep neural networks from a physical perspective. This process considers not only the relationship between different components within a sample point, but also interaction between sample pairs. The proposed method:

- promotes the discrete nature of sample points in a deep learning model.
- explicitly models the feature interaction between sample pairs.
- is demonstrated to improve the capacity of deep models.


**Summary Of The Review:**

Overall I think this paper provides an interesting perspective on feature post-processing for neural networks that considers both intra-sample and inter-sample feature interactions. However there is much room for improvement in the writing of the article, see the Weakness section above. I therefore tend to give a reject.

---

### Author Response · Authors · 2022-11-18
**General Response**

We thank all the reviewers for their precious time and carefully considered comments. We would like to respond to the shared questions among the reviewers here.

1. First of all, we deeply thank reviewers R-KJc9, R-hSD9, R-vH4E, R-vXfj for finding this work novel, interesting and surprising. We explore the connection between Atomic Physics and widely-used complex data with subunits (e.g., natural language and visual data). We experiment and observe the properties of our proposed method on smaller-scaled but various setups, a trade-off of our accessible computational resources, in synthetic, text, and image domains with different ranges of label numbers and data sizes. We summarized that the proposed Atom Modeling can help shape the representation when training a deep learning model, which further improves the performance and also gives an interpretation of the interactions among subunits.

2. We would like to clarify our motivations. We are motivated by the discrete nature of atoms. This discrete nature has not been considered carefully in the data that may be underlying discrete and have subunit embeddings. We anticipate that the property described in Section 2.1 is one reason that borrowing the idea of Atoms can benefit a deep learning model.

3. We sincerely thank all the reviewers for suggesting related work (R-KJc9, R-hSD9, and R-vH4E for contrastive learning, R-qaGR and R-vXfj for geometry learning, R-hSD9 and R-vH4E for disentanglement regularization). After carefully thinking, according to our interpretation, Atom Modeling is not in the same line as the suggested fields. The reason is that Atom Modeling (1) atomizes each data point without any supervision and data augmentation (compared to contrastive learning using class labels or data-dependent augmentation; center loss or OLE using class labels), (2) discretizes data points based on no specific geometry assumption (compared to geometry learning and hyperbolic representation with a hierarchical assumption), (3) is not about disentangling attributes in data from its representation but modeling data as atoms.

We will take into consideration all precious suggestions in our future version. Thank you again for telling us your thoughts.

---

### Decision · Program_Chairs · 2023-01-20

**Decision:**

Reject

**Justification For Why Not Higher Score:**

This paper simply did not meet the bar of acceptance, and all reviewers agreed.

**Justification For Why Not Lower Score:**

Can't be lower.

**Metareview: Summary, Strengths And Weaknesses:**

This paper presents an intriguing idea of mapping concepts from atomic structure into machine learning, targeted at modeling intra-example and inter-example relations through analogies of charges, electrons, protons and neutrons.

However, it is not clear from the paper why such new concepts would be helpful for the learning tasks.  All reviewers found the motivation for introducing these concepts lacking, and the experiment results unconvincing.

I hope the authors can take the suggestions and improve their paper.